# Revealing the Correlation between Molecular Structure and Corrosion Inhibition Characteristics of N-Heterocycles in Terms of Substituent Groups

**DOI:** 10.3390/ma16062148

**Published:** 2023-03-07

**Authors:** Li Tan, Jiusheng Li, Xiangqiong Zeng

**Affiliations:** 1Laboratory for Advanced Lubricating Materials, Shanghai Advanced Research Institute, Chinese Academy of Sciences, Shanghai 201210, China; 2University of Chinese Academy of Sciences, Beijing 100049, China

**Keywords:** N-heterocycles, corrosion inhibition, substituent

## Abstract

Controlling metal corrosion can directly address the waste of metal and the environmental pollution and resource depletion caused by metal recycling, very significant factors for green and sustainable development. The addition of corrosion inhibitors is a relatively cost-effective means of corrosion prevention. Among these, N-heterocycles have been widely used because heteroatoms contain lone pairs of electrons that can be strongly adsorbed onto metals, protecting them in highly corrosive environments at relatively low concentrations. However, due to the large variety of N-heterocycles, their corrosion inhibition characteristics have seldom been compared; therefore, the selection of appropriate N-heterocycles in the development of anti-corrosion products for specific applications was very difficult. This review systematically analyzed the influence of different substituents on the corrosion inhibition performance of N-heterocycles, including different alkyl chain substituents, electron-donating and electron-withdrawing substituents, and halogen atoms, respectively. The correlation between the molecular structure and corrosion inhibition characteristics of N-heterocycles was comprehensively revealed, and their action mechanism was analyzed deeply. In addition, the toxicity and biodegradability of N-heterocycles was briefly discussed. This study has provided a significant guideline for the development of green, promising corrosion inhibitors for advanced manufacturing and clean energy equipment protection.

## 1. Introduction

Metallic materials and alloys can be seen everywhere in production and life. Due to their irreplaceable and excellent mechanical properties, they are used in defense, construction, medicine, energy, transportation and other fields. However, the metals used in manufacturing and life, including iron, copper, aluminum, and others, are commonly confronted with a serious problem: namely, metal corrosion. Corrosion is defined as the chemical and electrochemical reaction of metals with certain specific components in the environment [1,2,3,4]. Metal corrosion can lead to product loss or contamination, production shutdown, toxic product leakage, and threats to life, health and environmental safety, etc. [5,6,7,8,9,10]. Therefore, the protection of metals is an important issue, especially with respect to advanced manufacturing and clean energy equipment protection under the Double Carbon strategy.

There are various methods for protection against metal corrosion. These methods generally include changing the metal itself to make an alloy, the cathodic protection method for the sacrificial anode, the applied current negative electrode protection method, the coating method, and the corrosion inhibitor method [7,8,9,10,11]. Among these, the inhibitor addition method is very popular under certain working conditions because of its convenient operation, low cost and high efficiency [12,13]. Currently, the types of corrosion inhibitors are complex and various, including inorganic corrosion inhibitors, organic corrosion inhibitors and complex corrosion inhibitors. The organic corrosion inhibitor has attracted a large number of researchers thanks to its structural diversity and maneuverability. With the enhancement of people’s awareness of environmental protection, scientists have gradually shifted the focus of their exploration to the use of “green”, pollution-free, non-toxic organic compounds as corrosion inhibitors such as amino acids, surfactants, polymers, ionic liquids, plants extracts and heterocyclic compounds containing active elements such as N, P, O, S, etc. [14]. Relatively speaking, because N-heterocyclic compounds are rich in variety, highly stable, low-toxicity and comply with environmental protection requirements and sustainable development strategies, they are widely researched. With increasingly strict environmental regulations, N-heterocyclic compounds serve as ideal substitutes, especially for natural sources and nontoxic types of compounds.

To date, there have been extensive studies on N-heterocyclic compounds applied in acidic and seawater corrosive environments. The most common corrosion inhibitors studied are pyrrole, pyrazole, imidazole, pyridine, pyridazine, pyrimidine, triazine, indole, quinoline and acridine [15,16,17,18,19], shown in Figure 1. In order to further improve the corrosion inhibition properties of N-heterocycles, researchers have generally modified N-heterocycles with substituents. By adding alkyl chains or polar groups to the parent, a synergistic interaction between the parent and the substituent is achieved to increase the corrosion inhibition properties of the compound. According to the literature, the most effective inhibitors are found to be N-heterocycles linked to functional groups such as alkyl, -NH_2_, -OCH_3_, -OH, -CHO, -CN, -COO-, -NO_2_, unsaturated (double bonds and triple bond) bonds and halogen atoms [20,21,22,23].

When N-heterocycles are used as corrosion inhibitors, they typically interact with the metal surface by electrostatic attraction or coordination links so that the corrosion inhibitor is adsorbed on the metal surface, affecting the corrosion potential, corrosion current and electric double-layer capacitance, thereby inhibiting or slowing down the metal corrosion rate. The existence of substituents will affect the electron distribution of the whole compound, thereby affecting the interaction and adsorption energy between the N-heterocyclic compound molecule and the metal and altering the inhibition performance of corrosion [16,17,18,19,24]. Therefore, the electronic and structural properties (geometry, charge and active site) of N-heterocycles are an important aspect in tailoring their corrosion inhibition properties [17].

The study of corrosion inhibitors must investigate the conformational relationship between the molecular structure and its corrosion inhibition ability at the molecular or atomic level and explain the mechanism of interaction between the corrosion inhibitor molecules and metal atoms. However, due to the large variety of N-heterocycles, their corrosion inhibition characteristics were seldom compared and analyzed. Therefore, in this paper, we reviewed and analyzed the influence of various substituents on the corrosion inhibition performance of N-heterocycles, including different alkyl chains, electron-donating and electron-withdrawing substituents and halogen atoms, respectively. The corrosion inhibition characteristics of the N-heterocyclic compound were revealed in detail, and their action mechanisms were comprehensively disclosed. Furthermore, the toxicity and biodegradability of N-heterocycles were briefly introduced, and the development of green corrosion inhibitors was considered. This paper will provide efficient guidelines and new ideas for the design of promising corrosion inhibitors for different environmental conditions.

## 2. The Correlation between Molecular Structure and Corrosion Inhibition Characteristics of N-Heterocycles

### 2.1. The Influence of Alkyl Chain Length

The alkyl chain has a great influence on the corrosion inhibition performance of the molecule. In some carboxylic acid derivatives, the length of the alkyl chain is a major corrosion inhibition factor [25,26]. Milošev showed that alkyl chains alone cannot be used as corrosion inhibitors, and carboxylic acid groups are needed as anchoring groups [27]. El-Haddad [28] explored the corrosion inhibition performance of imidazole derivatives with and without alkyl groups for aluminum in 0.5 M HCl. Their results revealed that the corrosion inhibition efficiency of methyl imidazole was better than that of imidazole. The methyl substituent did not affect the mechanism of action of the imidazoline but increased the coverage and adsorption of the compound on the aluminum surface. Additionally, the adsorption obeyed the Frumkin adsorption isotherm. In the study by M. Ontiveros-Rosales, it was found that the completely adsorbed IBPI3 structure was similar to that obtained by the cluster model, and that the cationic and anionic parts were T-shaped. The single N of the imidazole salt part was perpendicular to the iron surface, the mechanism of which is provided in Figure 2 [25]. Therefore, in this section, we focus on discussing N-heterocycles as anchoring groups and the effect of alkyl substituents with different chain lengths on the corrosion inhibition performance of N-heterocycles. The main chemical structures are shown in Table 1, in which the number of nitrogen atoms in the ring is sorted from less to more, that is, following the order of the five-membered ring, six-membered ring and fused-ring heterocycle. 

Guo and Kozaderov et al. [34] explored the corrosion inhibition behavior of triazoles with different alkyl chain lengths by the density functional theory. Electrons were distributed on the triazole ring, meaning that the preferred position for the electrophilic reaction of the compound was likely to be on the ring, and the adsorption was in a planar configuration. The absolute value of adsorption energy increased with the increase in alkyl chains, and the corresponding inhibition efficiency also followed the same law. Kozaderov [33] concluded that the longer the length of the triazole alkyl chain, the greater their inhibition effect for a copper corrosion system in a medium of sodium chloride with a multi-component passivation film formed on brass, including ZnO·xH_2_O, Cu_2_O and CuO oxides and a small number of soluble complexes of zinc and copper with the studied inhibitor molecules. Yoo et al. [37] synthesized a series of tris(carboxy alkylamino)triazines with varying lengths of the alkyl chain and found that they were mixed adsorption corrosion inhibitors that demonstrated a corrosion inhibition efficiency value that increased strongly with an increasing alkyl chain length. Hu et al. [38] synthesized BTC_6_T and BTC_8_T triazine compounds. It was revealed that their high inhibition efficiency was attributed to their adhesion and adsorption to prevent metal dissolution and hydrogen evolution reactions. When the alkyl chain length of the triazine derivatives was longer, the compounds had higher adsorption equilibrium constants and a lower Gibbs adsorption free energy. Therefore, the authors confirmed that the alkyl chain length plays an important role in the corrosion inhibition properties of triazine derivatives.

In addition to the modification of five-membered heterocycles such as imidazole, researchers are also optimistic about the corrosion inhibition performance of six-membered N-heterocycles. Han et al. [34] studied three pyridyl gemini surfactants with different alkyl chain lengths. The corrosion resistance was improved by increasing the length of the alkyl chain attached to the pyridine rings. Meanwhile, a contact angle test proved that the hydrophobicity was stronger when the alkyl chain was longer. Shaban et al. [36] also studied the corrosion inhibition performance of three pyridinium surfactants for API 5L X70 carbon steel in oilfield formation water and found that the critical micelle concentration value decreased with an increasing alkyl chain length. The corrosion properties were consistent with this. Feng et al. [35] studied the corrosion inhibition of carbon steel by pyridine derivatives with different alkyl chain lengths (n = 8, 10 and 12) in a 1 M HCl solution. However, it was found that at low concentrations, the carbon chain length of Geminis had a negligible effect on the corrosion inhibition efficiency.

Due to the hydrophobicity, N-heterocyclic corrosion inhibitors with long alkyl chains are somewhat limited in use due to their solubility. The advent of ionic liquids (ILs) has solved the solubility problem and become an excellent alternative in the field of corrosion inhibitors [42,43,44,45,46,47]. Ionic liquids are generally divided into a cationic part and an anionic part, and an alkyl group is generally used as a modified part of the cation. Feng et al. [24] studied three anionic corrosion inhibitors, 1-vinyl-3-methylimidazolium iodide ([VMIM]I), 1-vinyl-3-propane-1-vinyl imidazolium iodide ([VPIM]I) and 1-vinyl-3-butyl imidazolium iodide ([VBIM]I), in a 0.5 M sulfuric acid solution for X70 steel. Similarly, Qiang et al. [29] compared the corrosion resistance of ethyl, butyl, hexyl, and octyl-modified 3-position 1-allyl-imidazole bromide to copper. For the above-mentioned systems, the corrosion inhibition efficiency increased with an increase in the alkyl chain length. In Arelanes-Lozada’s study, the polarization resistance value (R_p_) indicated that PImC_12_, with the longest alkyl chain length, formed the most dense film on the surface. Compared with the aliphatic chain lengths of other PILs, the effective arrangement of PLmC_12_ had the greatest protection capability on the metal surface shown in Figure 3 [30]. Cao et al. [31] explored the inhibition efficiency of ILs ([(CH_2_)COOHMIm][HSO_4_], [(CH_2_)_2_COOHMIm][HSO_4_]) for carbon steel in a 0.5 M HCl solution. It was found that ionic liquids can improve the solubility of imidazole rings, and N-heterocyclic parent cations and anions can be adsorbed on metal surfaces as active adsorption sites. The inhibition efficiencies of the above ILs increased with the augment of the alkyl chain length attached to the imidazolium ring, and their favorable performance was ascribed to the formation of IL adsorption films. Moreover, due to the covalent bond formed by charge transfer or sharing and electrostatic attraction, ILs were mainly chemisorbed onto the metallic surface. The longer alkyl chain yielded a stronger interaction of the studied ILs and a larger coverage area on the metal, forming a denser protective film to reduce aggressive attacks.

In order to reveal the action mechanism at the molecular level, the molecular dynamic simulation method was adapted to simulate a large number of N-heterocycles with different alkyl chain lengths. The corrosion inhibition efficiency was evaluated according to the diffusion coefficient of corrosion ions in the corrosion inhibitor film. Three related parameters, that is, the free volume fraction of the film, the interaction energy between the film and the particles and the mobility of the corrosion inhibitor film, were discussed to clarify the diffusion behavior of the corrosive particles in the inhibitor film. Yan [19] and Yu et al. [18] found that the parent part of the N-heterocycle is the active adsorption site adsorbed on the metal surface. The organic alkyl chain deviated from the metal surface and achieved stable adsorption through its torsional deformation. This conclusion also supported the previous finding that the single alkyl chain could not act as an adsorption corrosion inhibitor. When the alkyl chain was short, its flexibility was low, and bending was difficult. The alkyl chain was inserted side by side into the corrosive medium due to its rigidity, and they hardly intertwined with each other [48]. At this time, the inhibitor film formed many large cavities and had a large free volume fraction. As the alkyl chain length increased, the increased flexibility promoted the intertwining of the alkyl chains, which divided these large cavities into many small cavities. At this time, the free volume fraction decreased. When the N-heterocyclic had a longer alkyl chain, it was easier to form a dense corrosion inhibition film, and the voids between the films were rare. In addition, the membrane had a strong interaction with the corrosive ions. The penetration of the corrosive ions between the adsorption membranes was difficult, and the diffusion probability and rate of the corrosive substances were small. Under dynamic equilibrium conditions, the corrosion inhibitor molecules themselves migrated and interacted due to van der Waals and Coulomb forces, and an increase in the length of the alkyl chain caused the self-diffusion coefficient to decrease dramatically. Therefore, under the double diffusion of a corrosion inhibitor and a corrosive medium, the diffusion of the corrosive medium to the metal surface can be effectively hindered, thus achieving the purpose of delaying metal corrosion.

In conclusion, the alkyl chain length has a great influence on the corrosion inhibition performance of the N-heterocyclic parent. In most cases, the longer the alkyl chain length, the stronger the corrosion inhibition capability of the system. The action mechanism is summarized below:(1)When the active group is adsorbed on the metal surface, the longer alkyl chain can increase the coverage and compactness of the corrosion inhibition film. At this time, there are few cavities between the corrosion inhibition films, and the free volume fraction of the corrosion substance is small, slowing down the diffusion coefficient of corrosive ions and increase the corrosion inhibition ability;(2)Due to its electron-donating characteristic, the alkyl chain can change the electron density of the parent body, thereby changing the overall adsorption energy. Therefore, the longer alkyl chain can increase the corrosion inhibitor’s interaction energy with metals, resulting in better corrosion inhibition performance;(3)The longer alkyl chain has greater hydrophobicity, preventing the approach of corrosive ions in water.

However, due to the type of corrosive medium, temperature, pH, and protective metal, the corrosion inhibition performance of precursors with different chain lengths varies greatly. Even in some specific cases, the alkyl chain length has little effect on the corrosion inhibition properties of the parent. We still need to analyze the situation on a case-by-case basis.

### 2.2. The Influence of Electron-Withdrawing and Electron-Donating Groups 

In most cases, corrosion inhibitors protect the metal by adsorption. In general, N-heterocyclic compounds use the N atom as an active center to form electron sharing or to coordinate lone-pair electrons with empty metal orbitals to achieve a layer of adsorption film for metal protection. The electron-donating and electron-withdrawing abilities of the attached substituent can change the overall electron distribution of the N-heterocycle, thereby affecting the adsorption performance and further affecting the corrosion inhibition ability. It is worth mentioning that whether a group is an electron-withdrawing group or an electron-donating group depends on the sum of its inductive effect, conjugation effect and hyperconjugation effect on the ring. Common electron-withdrawing groups are -N^+^R_3_, -NO_2_, trihalomethane (-CX_3_, X=F, Cl), -CN, -SO_3_H, -CHO, -COR and -COOH. Electron-donating groups are -O-, -NR_2_, -NHR, -NH_2_, -OH, -OR, -NHCOR, -OCOR, -R, -CH_2_COOH and -Ph. In the next section, we discuss the effect of electron-withdrawing and electron-donating substituents on the corrosion inhibition performance of N-heterocyclics. Examples of molecular structures are shown in Table 2.

Ouakki et al. [64] studied in detail the effect of electron-withdrawing and electron-donating groups on the corrosion inhibition performance of imidazole rings and used the Hammett substituent constant (σ) to describe the electronic effects of the substituents. The Hammett substituent constants of various substituent groups are summarized in Table 3 in which σ_m_ represents the meta substituent constant, and σ_p_ represents the para substituent constant. Generally, the smaller the σ, the stronger the inhibition effect, which represents a stronger adsorption and higher surface coverage [20,21].

For five-membered N-heterocycles, El-Haddad et al. [28] found that as an electron-donating group, the methyl group could increase the corrosion inhibition performance of the N-heterocycle parent. Gutiérrez [50] analyzed a set of imidazole and benzimidazole derivatives and confirmed that the properties such as electronegativity, aromaticity, volume and nitrogen atomic charge of the heterocycles are strongly related to resistance to corrosion while also confirming that the presence of halogens with an electron-withdrawing ability could improve the corrosion inhibition efficiency. Similarly, Zhang [51] showed that the corrosion inhibition efficiency of 1-R-2-undecyl imidazoline compounds was in the order of CH_2_COOH (A) > CH_2_CH_2_OH (B) > CH_2_CH_2_NH_2_ (C) > H (D)) for carbon steel, which is consistent with the decrease sequential of the Hammett substituent constants as exhibited in Table 3, suggesting that the stronger the electron-withdrawing ability of the group connected to the parent, the stronger the corrosion inhibition performance of the entire N-heterocyclic molecule. Additionally, Cai et al. [52] investigated three imidazole-based ionic liquids substituted by polar groups (-COO^−^,-Ph) at N1 in 0.3 M NaCl-saturated Ca(OH)_2_ solution. It was found that the corrosion inhibition effect of the carboxyl substituent (electron-withdrawing group) was better than that of the phenyl substituent (electron donating group).

Regarding pyridine derivatives, Tebbji [55] studied the corrosion inhibition effect of pyridine-pyrazoles(P1, P2) with different substituents(-CH_2_OH and -COOC_2_H_5_) in HCl for steel. The high inhibition efficiency was explained by the presence of three nitrogen atoms, and electrons of the pyridine and pyrazole ring were the major adsorption centers for their interaction with the metal surface [65]. The difference was attributed to the presence of an electron-donating group (CH_2_OH) of P1 and an electron-withdrawing group (COOC_2_H_5_) of P2 by the mesomeric effect to change the density of the molecule. It was found that the substituents did not significantly change the corrosion inhibition mechanism of the main body but did affect the corrosion inhibition efficiency (P1 > P2). Ansari [43] evaluated the influence of -H, -CH_3_ and -OCH_3_ on the corrosion inhibition performance of compound PC, finding that the electron-donating group reduced the corrosion resistance of the compound instead. It is worth mentioning that the adsorption on the solid surface is not purely physical or chemical, as illustrated in Figure 4. First, the chemical adsorption of PCs arose from the donor–acceptor interactions between free electron pairs of heteroatoms and π-electrons of multiple bonds as well as the phenyl group and vacant d-orbitals of irons [66,67]. The second was physical adsorption from charged metal surfaces to charged PC molecules. Furthermore, the adsorption of chloride ions on the positively charged metal surface generated an excess negative charge, and the positively charged (PCsH^+^) adsorbed on the metal surface through electrostatic interaction and formed a protective layer [68,69]. Finally, to release the extra negative charge from the mild steel, the electrons from the d orbital of Fe may be transferred to the empty p* orbital of the PC molecule, thereby enhancing the adsorption on the surface of the mild steel. The above adsorption was closely related to the overall electronic structure of the inhibitor molecule.

A six-membered heterocyclic pyrazine containing two N atoms in the para position (AP and ABPT) was investigated by Saha et al. [47]. Based on the molecular dynamic simulation, it was also found that the compound forms coordinated through the donor–acceptor interaction between the lone-pair electrons on N and the π electrons on the pyrazine ring with the vacant d orbital of Fe. At the same time, there might also be chelation with Fe^2+^.

Among the quinoline derivatives, Wang [59] studied the effect of the electron-donating amino group and the electron-withdrawing nitro group on their corrosion inhibition properties. For 8-AQ, its interaction with Al (1 1 1) was through the amino group (Figure 5a), while the interaction between 8-NQ and Al (1 1 1) was through the O atom of the nitro group (Figure 5b). The corrosion inhibition efficiency was 8-NQ > 8-AQ. From the DFT calculation, the electron-donating group (-NH_2_) allowed the charge accumulation region to be biased towards Al atoms, while 8-NQ, with the electron-withdrawing group (-NO_2_), had a bias towards O atoms, which were N and O atoms that provided lone pairs of electrons to form coordination bonds with Al atoms. The above adsorption was performed by the substituent connected to the parent; therefore, the substituent played a vital role in the corrosion inhibition process of the compound. 

Regarding benzimidazole derivatives, in the study by Aljourani [56], the thermodynamic adsorption parameters indicated that 2-mercapto benzimidazole, 2-methyl benzimidazole and benzimidazole retarded the cathodic and anodic processes by physically adsorbing and blocking active corrosion sites on mild steel in 1 M HCl. The corrosion inhibition efficiency was in the order of 2-mercaptobenzimidazole > 2-methyl benzimidazole > benzimidazole. Combined with the electron-donating ability of the substituents, it was found that stronger electron-donating substituents can improve the overall corrosion inhibition efficiency of the N-heterocycle. Similarly, Ghanbari [70] studied the corrosion inhibition ability of benzimidazole (BI), 2-methyl benzimidazole (2MBI) and 2-amino benzimidazole (2ABI) for mild steel in 1 M phosphoric acid and found that their inhibitory properties were in the order of 2ABI > BI > 2MBI. By using the Flory–Huggins adsorption isotherm model, it was revealed that each inhibitor replaced 3–5 molecules of water on the metal surface. Different from Aljourani’s conclusion, the corrosion inhibition performance of benzimidazole was better than that of 2-methyl benzimidazole because the external environment, such as the corrosive medium, had a great influence on the corrosion inhibition ability of the compound.

The above studies proved that electron-withdrawing and electron-donating groups have a great influence on the corrosion inhibition performance of the N-heterocyclic matrix, although there is no clear correspondence between the electron-donating or electron-withdrawing substituent and the corrosion inhibition ability of N-heterocycle. The law of influence was not fixed and is sometimes even controversial. The action mechanism is summarized below:(1)Electron-donating and electron-withdrawing groups affect the adsorption capacity by changing the overall electron density. When the adsorption energy and adsorption area are reduced, the corrosion inhibition ability is also reduced;(2)When the molecular volume of the N-heterocycles is large or very small, the presence of the substituent was negligible for changing the electron distribution of the entire molecule. The addition of electron-donating and electron-withdrawing groups directly changes the number of adsorption sites, at which point the electron-rich heteroatomic centers are directly involved in binding to the metal surface. These electron-rich centers transfer their non-bonded and π electrons to the d orbitals of atoms on the metal surface, forming coordination bonds. At this time, both the electron donating and electron-withdrawing groups increase the corrosion inhibition performance of the compound;(3)In addition, when the substituent changes the hydrophobicity of the entire molecule, the corrosion inhibition efficiency will be affected as well due to the change in solubility of the N-heterocycles. On the one hand, highly hydrophilic compounds may be solvated and cannot adsorb on metal surfaces. On the other hand, highly hydrophobic compounds precipitate due to insoluble polar electrolytes. Both high hydrophilicity and high hydrophobicity will reduce the corrosion inhibition efficiency of the compounds;(4)Besides, the corrosive medium and material of the metal also have a great influence on the corrosion inhibition effect of the compound; therefore, the effect should be discussed according to the specific situation of practical application.

### 2.3. The Influence of Halogen Substituent Group

In general, halogen atoms are a type of electron-withdrawing group; however, they belong to a specific column in the periodic table. For this purpose, we discussed in detail the effects of halogen substituents on the corrosion inhibition behavior of N-heterocycles. Their structural formulas are shown in Table 4.

The adsorption behavior and corrosion inhibition mechanism of C2-halogenated derivatives of imidazole (2-I-Imz, 2-BrImz and 2-Cl-Imz) on an Fe(100) surface were studied by Ismail Abdulazeez et al. [71]. Electrochemical tests demonstrated that the steady-state corrosion potential was in the order of 2-I-Imz > 2-Br-Imz > 2-Cl-Imz, with the adsorption energies of the Fe(100) surface being −3.98, −3.76 and −3.48 eV, respectively, indicating that the thermodynamic stability of the surface film and the corrosion inhibition properties increased following the same order. This was attributed to the ability of the dissociated halogen substituents to bind to the Fe^2+^ on mild steel. As the concentration increased, the coverage of the iron surface increased. The facile dissociation and strong adsorption of 2-I-Imz suggested its potential as an excellent corrosion inhibitor relative to imidazoles and other halogenated imidazoles. During the adsorption process, the parallel conformation of the molecule was more stable due to the large adsorption area with the metal, and the inclined conformation was kinetically unstable in Figure 6, mainly because the C-halogen bond was easily broken. The dissociated product was strongly bound to the iron surface, thus enabling the formation of a strong adsorption layer with properties that depended on the halogen–metal interaction. However, the dissociated Cl^−^ and Br^−^ corroded by exchanging with the oxygen in the metal-protective oxide film. More pitting damage occurred on the metal surface, with chloride showing the most damage.

Chaitra et al. [76] combined electrochemical experiments and theoretical calculations to explain the anti-corrosion mechanism of triazole derivatives (BFBT, TMBT, and FNBT). The corrosion inhibition efficiency for mild steel in 0.5 M HCl could be explained in terms of the number of adsorption sites, molecular size and the way they interact with the metal surface. The inhibitors contained three nitrogen atoms, an amide bond and π electrons as adsorption centers. Therefore, the inhibitor was chemisorbed to the metal surface via these electrons through coordination bonds. The nitrogen atoms were easily protonated and physically bonded to positively charged mild steel surfaces via negatively charged chloride ions (Cl^−^). Bromine atoms had isolated pairs of electrons that were capable of transmitting them to the d orbital of the Fe and provided the highest inhibition efficiency of BFBT [77]. The FNBT exhibited the lowest inhibition efficiency due to the presence of the strongest passivating groups such as nitro, which reduced the electron density [78]. This demonstrated that the aforementioned substituents affect the electron cloud density of the parent and affect the adsorption between the metal and the corrosion inhibition performance. Li et al. [72] also studied the corrosion inhibition performance of halogen-substituted triazoles and found their polarization and impedance were exhibited in the order of CATM(-Cl) > MATM(-OCH_3_) ≈ FA TM(-F). As shown in Figure 7, protonated triazole was indirectly adsorbed on the surface of mild steel through the synergistic effect with chloride ions to inhibit corrosion [79,80,81]. The presence of substituents directly modifies its adsorptive energy.

Vidhya [73] explained the corrosion inhibition performance of halogen-substituted pyridine derivatives through the DFT and the electron charge on the chelating atom. Compared with 2-bromopyridine, 2-chloropyridine had the highest HOMO energy (E_HOMO_) and the lowest LUMO energy (E_LUMO_). The presence of halogens increased the charge on the active atoms of the compound. Sorkhabi et al. [46] investigated the corrosion inhibition performance of benzylidene-pyridine-2-yl-amine (A), (4-benzylidene)-pyridine-2-ylamine (B) and (4-chloro-benzylidene) -pyridine-2-yl-amine (C) for mild steel in 1 M HCl. The double-layer capacitance decreased in the impedance test, proving the adsorption of compound molecules on the surface of carbon steel. The calculation results showed that in the structure of the compound, the benzene ring atom and the -C=N group formed a large π bond. Not only did the π electrons of the Schiff base enter the empty orbital of the iron, but the π orbital also accepted the electrons of the d orbital of the iron to form a feedback bond, generating more than one chemisorption center and increasing the inhibition efficiency.

In addition to substituting single heterocycles, halogens will also replace fused heterocycles. Fernandes et al. [75] synthesized three halogenated quinoline derivatives: DHOP-X (X = F, Cl, Br). Due to the decrease in electronegativity from fluorine to bromine atoms, the larger electron π density increased correspondingly along the series, providing a higher corrosion protection efficiency. This trend was confirmed by DFT calculations of several parameters related to the molecular electronic structure, including HOMO and LUMO energies, electronegativity, bulk hardness, softness and dipole moment.

In summary, the action mechanism of halogen atoms on the overall corrosion inhibition performance of N-heterocycles is mainly through the electronic effect of halogen substituents. As a rule, the electronegativity of halogen elements refers to the ability of halogen atoms to attract electrons. The larger the electronegativity value of the halogen atom, the stronger the ability of the halogen atom to attract electrons in the compound. In the periodic table of elements, the electronegativity of the halogen elements grows increasingly weak from top to bottom, following F > Cl > Br > I. In general, the corrosion resistance of halogenated compounds is negatively correlated with electronegativity, and their corrosion inhibition properties are generally -I > -Br > -Cl > -F. Taking fluorine substituents as an example: fluorine substituents have a strong electron-withdrawing ability, and the presence of fluorine atoms will consume electron density from the molecular π system, thereby reducing the adsorption activity of adsorption sites. The values of the parameters related to the electronic structure change, such as the decrease in E_HOMO_, the increase in the E_LUMO_ value, the increase in the energy band gap (ΔE) and the increase in the dipole moment, finally affecting the adsorption ability of the N-heterocycles on the metal surface. The reduced adsorption capacity reduces the coverage on the metal surface, resulting in less protection for the metal. Moreover, the corrosion inhibition performance is related to the stability of the substituents. As the C-halogen bond is easily broken, the dissociated Cl^−^ and Br^−^ may attack the metal surface as corrosive species.

It is well known that most N-heterocycles with corrosion inhibitory effects in acid or salt corrosion environments are physical, chemical or mixed adsorption corrosion inhibitors. The adsorption mode of corrosion inhibitors is usually determined by criteria based on the standard adsorption free energy. Values greater than −20 kJ/mol are attributed to physical adsorption, and values less than −40 kJ/mol are attributed to chemisorption, which has been questioned by scientists [82]. However, we found in this paper that the corrosion inhibition properties of compounds are directly related to the Gibbs free energy. The higher the corrosion inhibition efficiency, the lower the Gibbs free energy value. Similarly, the longer the alkyl chain, the lower the Gibbs free energy. The Gibbs free energy of the corrosion inhibitors presented in the paper is shown in Table 5.

## 3. The Toxicity and Biodegradability of N-Heterocycles

Due to their high corrosion inhibition ability, N-heterocyclic compounds have been used as effective corrosion inhibitors for different metals and alloys. However, with the increasingly strict environmental protection laws and regulations, in order to strengthen ecological environment protection, the use of corrosion inhibitors is becoming more and more strict. Therefore, it is worth paying attention to the toxicity and biodegradability of N-heterocycles.

Some nitrogen heterocyclic organic compounds are highly toxic and refractory substances which seriously harm organisms. For example, aromatic N-heterocycles are generally considered to have the “three causes” effects of teratogenicity, carcinogenicity and mutagenesis, which may directly lead to the death of organisms or may exist in organisms to cause genetic mutations of the organisms or flow in the ecological chain, causing wider harm [83,84,85,86,87,88]. In the case of pyridine [89], exposure to pyridine compounds can cause nose and skin irritation. Exposure to high concentrations of pyridine may cause unconsciousness and death. Exposure to low concentrations may cause narcotic effects, including nausea, mental depression, anorexia and fatigue. Therefore, we should avoid developing and using aromatic N-heterocycles, pyridine derivatives, etc.

However, there are few studies on the toxicity and biodegradability of N-heterocycles. Vryzas [90] and Peddinghaus [91] et al. assessed the embryotoxicity of heterocyclic compounds in zebrafish using an assay technique. High mobility and persistence make the release of these compounds a potential hazard in aquatic environments.

In order to ensure green and sustainable development, it is important to conduct toxicity and biodegradability studies when developing N-heterocyclic corrosion inhibitors. In the meanwhile, a green and cost-effective synthetic methodology must be considered as well.

## 4. Conclusions and Outlook

In this paper, the inhibitory properties of N-heterocycles with different substituents were reviewed. Corrosion inhibition depends on many factors, including the molecular structure of the inhibitor itself, the number of adsorption sites, the protective metal, the corrosive medium and the ambient temperature. N-heterocycles exhibit different inhibition efficiencies for different metals in different media. All inhibitors prevent corrosion by acting as a barrier layer between the substrate and the environment. Among them, different substituents control the overall corrosion inhibition ability by changing the original hydrophobicity, solubility, electronic structure, molecular volume and active sites of the compound. From our summary and analysis, it can be concluded that:(1)Under the premise of solubility, the longer the alkyl chain, the stronger the corrosion inhibition ability of N-heterocycles. The alkyl substituent mainly affects the overall anti-corrosion ability by controlling the coverage, density and cavity of the corrosion protection film formed, the adsorption energy and the diffusion coefficient of corrosive ions. Long-chain alkyl substituents with high hydrophobicity increase the distance and difficulty for the corrosive medium to reach the metal;(2)The effect of electron-donating groups and electron-withdrawing groups on the corrosion resistance of N-heterocycles cannot be generalized. In some cases, both electron-donating and electron-withdrawing groups can increase the corrosion inhibition ability of the compound. The electron-donating and electron-withdrawing substituents mainly change the electronic structure of the N-heterocycle and the active site for adsorption, thereby changing the adsorption capacity of the compound and affecting the formation of an adsorption film and the blocking of corrosive media;(3)The influence of a halogen substituent mainly depends on its electronegativity. The overall corrosion inhibition ability of N-heterocycles is negatively correlated with the electronegativity of halogen substituent. The halogen substituents will consume the electron density of N-heterocycles, thereby affecting the overall electron cloud and adsorption capacity, and the dissociated halogen ions may become a new corrosion medium and cause pitting corrosion.

The above analyzed correlations between molecular structure and corrosion inhibition characteristics can act as useful guidelines for the development of new promising N-heterocyclic corrosion inhibitors, especially the study of imidazole, pyridine and triazole compounds. It is believed that an increasing number of N-heterocyclic compounds will be developed and applied because of their advantages in the field of corrosion inhibitors. While in practical research, due to the combined effect of various factors and the specific test conditions, we suggest that the characteristics of each compound be carefully revealed with a combination of experimental tests and computational methods. In situ novel techniques based on the Synchrotron Radiation Facility, such as micro-IR, XANES, etc., and effective theoretical calculation methodologies must be explored to deeply disclose the action mechanism. In theoretical calculation and molecular simulation, there are few studies on the bond length between corrosion inhibitors and metals; therefore, specific calculated values can be added to aid in the effective selection of functional groups. At the same time, molecular structures containing N-heterocyclic can be adjusted according to the corrosion environment to find the most appropriate corrosion inhibitor molecule. In addition, the toxicity and biodegradability of N-containing heterocycles are very important aspects to be considered as well.

## Figures and Tables

**Figure 1 materials-16-02148-f001:**
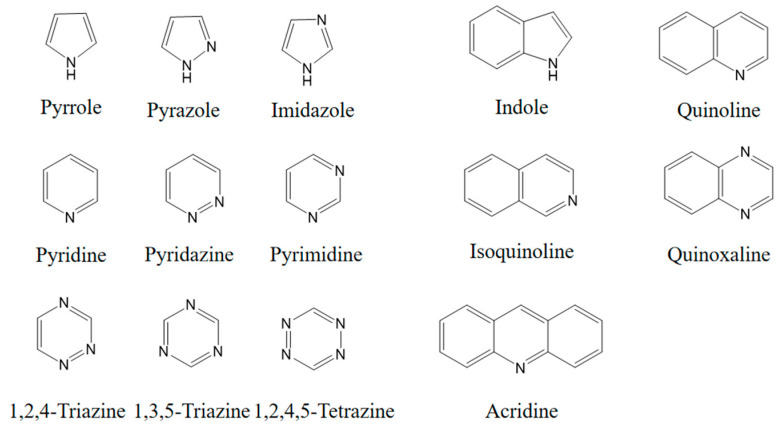
Illustration of the molecular structure of commonly used N-heterocycles.

**Figure 2 materials-16-02148-f002:**
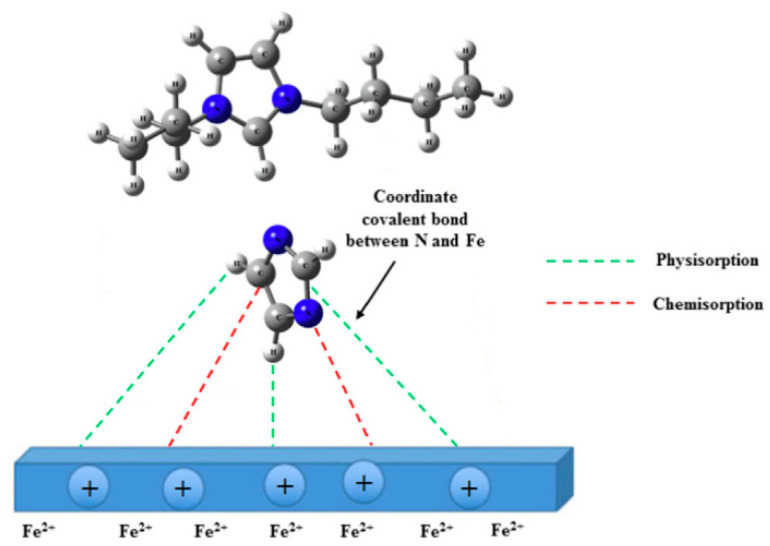
IBEI3 Schematic diagram on steel surface at a saturated CO_2_ concentration of 3% (*m*/*v*) [25].

**Figure 3 materials-16-02148-f003:**
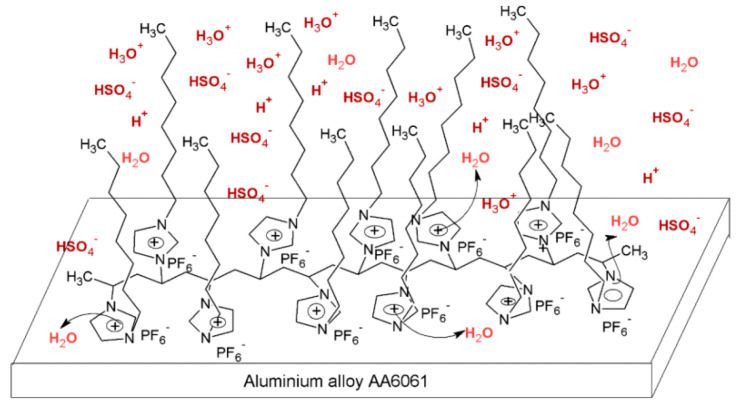
PIL adsorption diagram [30].

**Figure 4 materials-16-02148-f004:**
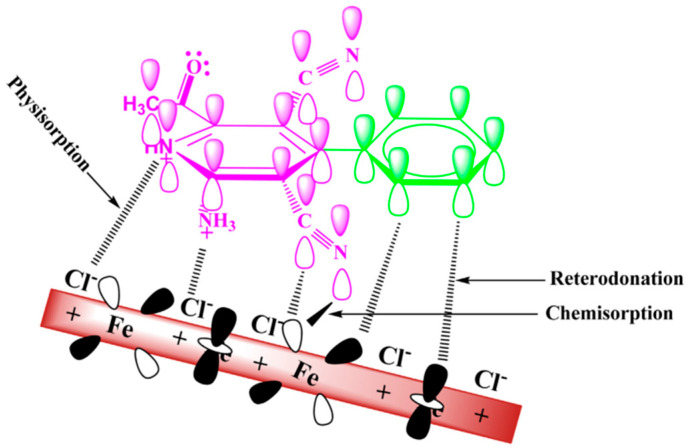
Schematic of adsorption and inhibition mechanism of PC molecule on mild steel in 1 M HCl [43].

**Figure 5 materials-16-02148-f005:**
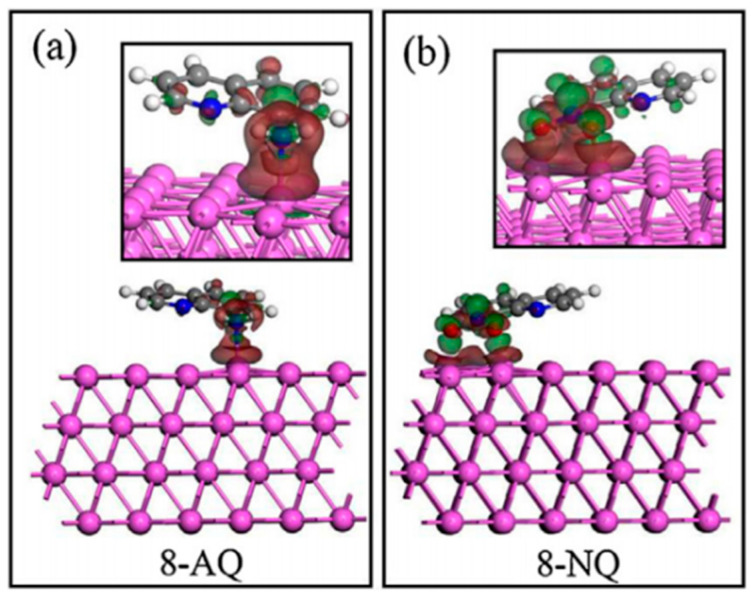
Charge density difference of inhibitor molecules absorbed onto Al(1 1 1): (**a**) 8-AQ; (**b**) 8-NQ [59].

**Figure 6 materials-16-02148-f006:**
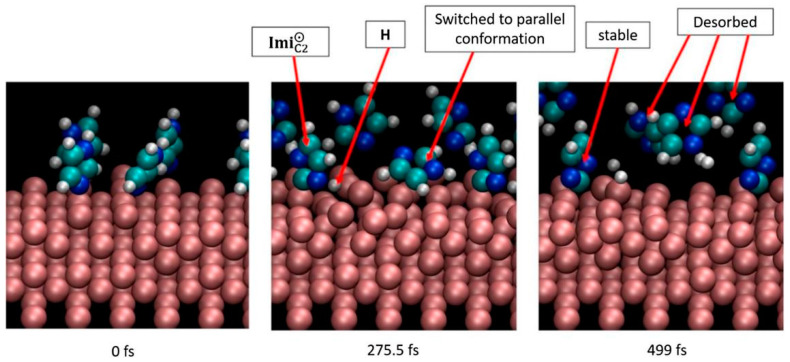
Imidazole molecules on an Fe(1 0 0) slab at 1500 K [71].

**Figure 7 materials-16-02148-f007:**
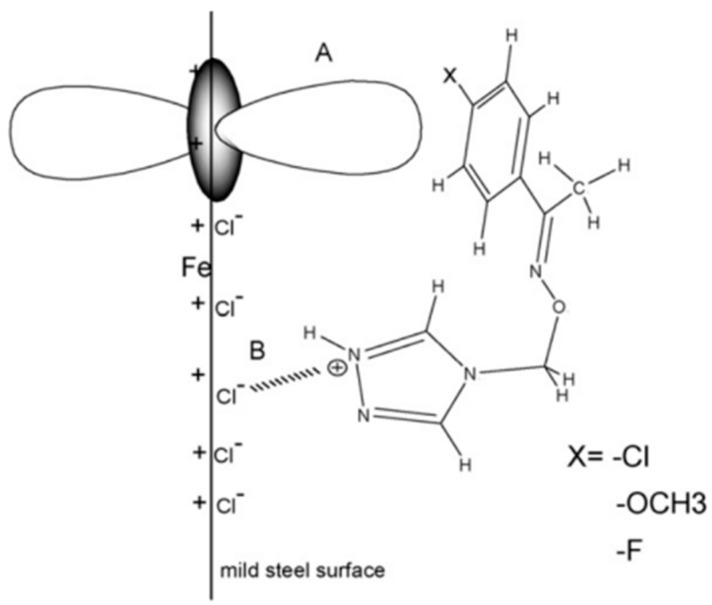
The schematic illustration of the adsorption behavior of the triazole derivatives on mild steel in 1 M HCl solution: (**A**) chemisorption and (**B**) physisorption [72].

**Table 1 materials-16-02148-t001:** Structural formulas of N-heterocycles with different alkyl chain lengths.

S No.	Chemical Structure/Abbreviation	Substituent	Adsorption Behavior	Ref.(s)
1	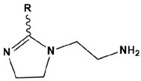	A:R=(CH_2_)_6_CH_3_B:R=(CH_2_)_8_CH_3_C:R=(CH_2_)_10_CH_3_D:R=(CH_2_)_12_CH_3_E:R=(CH_2_)_14_CH_3F_F:R=(CH_2_)_16_CH_3_G:R=(CH_2_)_18_CH_3_ H:R=(CH_2_)_20_CH_3_	-	[18]
2	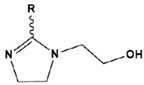	IC-7:R=(CH_2_)_6_CH_3_IC-9:R=(CH_2_)_8_CH_3_IC-11:R=(CH_2_)_10_CH_3_IC-13:R=(CH_2_)_12_CH_3_IC-15:R=(CH_2_)_14_CH_3F_IC-17:R=(CH_2_)_6_CH_3_	-	[19]
3	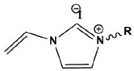	[VMIM]I: R=CH_3_[VPIM]I: R=(CH_2_)_2_CH_3_[VBIM]I: R=(CH_2_)_3_CH_3_	Mixed-type inhibitors/Langmuir	[24]
4	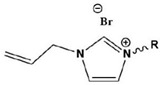	[AEIM]Br: R=CH_2_CH_3_[ABIM]Br: R=(CH_2_)_3_CH_3_[AHIM]Br: R=(CH_2_)_5_CH_3_[AOIM]Br: R=(CH_2_)_7_CH_3_	Cathodic inhibitors/Langmuir	[29]
5	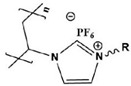	PImC_4_:R=(CH_2_)_3_CH_3_PimC_8_:R=(CH_2_)_7_CH_3_PimC_12_:R=(CH_2_)_11_CH_3_	Langmuir	[30]
6	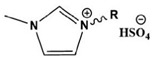	A:R=CH_2_COOHB:R=(CH_2_)_2_COOHC:R=(CH_2_)_3_SO_3_H	Langmuir	[31]
7	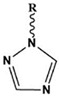	A:R=(CH_2_)_3_CH_3_B:R=(CH_2_)_6_CH_3_C:R=(CH_2_)_9_CH_3_	-	[32]
8	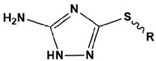	A: R=(CH_2_)_2_CH_3_B: R=(CH_2_)_3_CH_3_	-	[33]
9	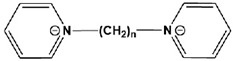	HBP:n = 6OBP:n = 8DBP:n = 10	Mixed-type inhibitors/Langmuir	[34]
10	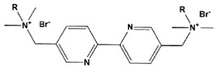	8-Bpy: R=(CH_2_)_7_CH_3_10-Bpy: R=(CH_2_)_9_CH_3_12-Bpy: R=(CH_2_)_11_CH_3_	Mixed-type inhibitors/Langmuir	[35]
11	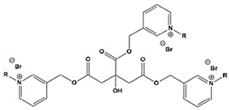	APMC6:R=(CH_2_)_5_CH_3_APMC12:R=(CH_2_)_11_CH_3_APMC18:R=(CH_2_)1_7_CH_3_	Mixed-type inhibitors/Langmuir	[36]
12	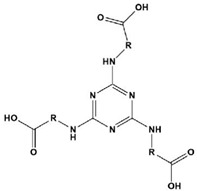	Tris-C1:R=(CH_2_)_1_Tris-C2:R=(CH_2_)_2_Tris-C3:R=(CH_2_)_3_Tris-C4:R=(CH_2_)_4_Tris-C5:R=(CH_2_)_5_	Mixed-type inhibitors/Langmuir	[37]
13	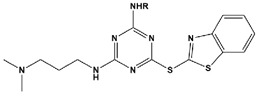	BTC_6_T: R=(CH_2_)_5_CH_3_BTC_8_T: R=(CH_2_)_7_CH_3_	Mixed-type inhibitors/Langmuir	[38]
14	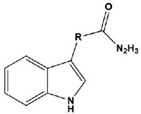	IAH: R=(CH_2_)_1_IBH: R=(CH_2_)_3_	Mixed-type inhibitors/Langmuir	[39]
15	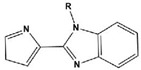	C1:R=(CH_2_)_1_CH_3_C2:R=(CH_2_)_3_CH_3_C3:R=(CH_2_)_5_CH_3_C4:R=(CH_2_)_7_CH_3_C5:R=(CH_2_)_9_CH_3_C6:R=(CH_2_)_11_CH_3_	-	[40]
16	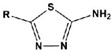	IC-2:R=(CH_2_)_1_CH_3_IC-3:R=(CH_2_)_2_CH_3_IC-5:R=(CH_2_)_4_CH_3_IC-7:R=(CH_2_)_6_CH_3_IC-11:R=(CH_2_)_10_CH_3_IC-13:R=(CH_2_)_12_CH_3_	Langmuir	[41]

**Table 2 materials-16-02148-t002:** Structural formulas of N-heterocycles with different electron-withdrawing and electron-donating groups.

S No.	Chemical Structure/Abbreviation	Substituent	Adsorption Behavior	Ref.(s)
1	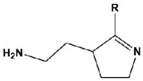	1-IM: R=CF_3_2-IM: R=CCl_3_	Mixed-type inhibitors/Langmuir	[49]
2	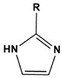	A: R=ClB: R=CH_3_	Mixed-type inhibitors/Langmuir	[50]
3	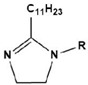	A:R=CH_2_COOH B:R=CH_2_CH_2_OH C:R=CH_2_CH_2_NH_2_D:R=H	-	[51]
4	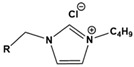	Comp. a: R=COO^−^ Comp.c:R=Ph	Langmuir	[52]
5	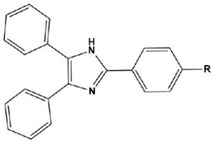	M1:R=OCH_3_M2:R=CH_3_M3:R=NO_2_	Mixed-type inhibitors/Langmuir	[53]
6	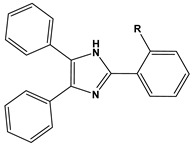	P1:R=HP2:R=OH	Mixed-type inhibitors/Langmuir	[26]
7	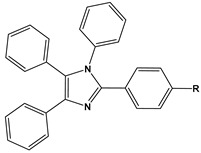	IM-Cl: R=ClIM-CH3:R=CH_3_	Mixed-type inhibitors/Langmuir	[54]
8	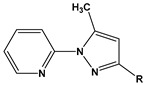	P1:R=CH_2_OHP2:R=COOC_2_H_5_	Frumkin	[55]
9	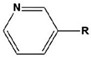	3-MP: R=CH_3_3-NM: R=NO2	-	[42]
10	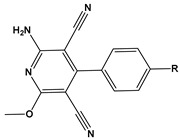	PC-1:R=HPC-2:R=CH_3_PC-3:R=OCH_3_	Mixed-type inhibitors/Langmuir	[43]
11	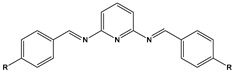	DAP-1:R=MeDAP-2:R=H	Anodic-type/Langmuir	[44]
12	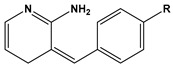	A: R=HB: R=CH_3_C: R=Cl	Donor-acceptor interaction	[45,46]
13	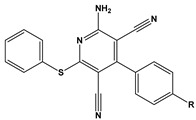	ADTP1:R=OCH_3_ADTP2:R=HADTP3:R=NO_2_	Mixed-type inhibitors/Langmuir	[43]
14	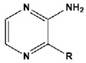	AP: R=HABPT: R=SH	-	[47]
15	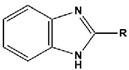	BI: R=H2-CH_3_-BI: R=CH_3_2-SH-BI: R=SH	Mixed-type inhibitors/Langmuir	[56]
16	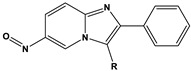	C1:R=CHOC2:R=CH_2_OH	Mixed-type inhibitors/Langmuir	[57]
17	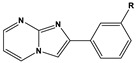	P1:R=HP2:R=OCH_3_	Mixed-type inhibitors/Langmuir	[58]
18	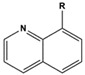	8-AQ: R=NH_2_8-NQ:R=NO_2_	Anodic inhibitors	[59]
19	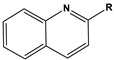	QL: R=HQLD: R=CH_3_QLDA: R=COOH	Langmuir	[60]
20	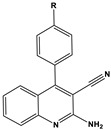	AAC-1:R=NO_2_AAC-2:R=HAAC-3:R=OH	Mixed-type inhibitors/Langmuir	[61]
21	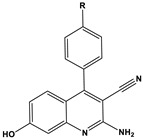	Q1:R=HQ2:R=CH_3_Q3:R=OCH_3_Q4:R=N(CH_3_)_2_	-	[62]
22	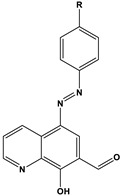	HL1:R=CH_3_HL2:R=HHL3:R=NO_2_	Mixed-type inhibitors/Langmuir	[63]

**Table 3 materials-16-02148-t003:** Values of some Hammett substituent constants (σ) [64].

Substituent	σ_m_	σ_p_
-CH_3_	−0.07	−0.17
-CH_2_CH_3_	−0.07	−0.15
-CH(CH_3_)_2_	−0.07	−0.15
-C(CH_3_)_3_	−0.10	−0.20
-CN	+0.56	+0.66
-COOH	+0.36	+0.43
-CHO	+0.36	+0.22
-CONH_2_	+0.28	+0.36
-CF_3_	+0.43	+0.54
-NH_2_	−0.16	−0.66
-NMe_2_	−0.15	−0.83
-NO_2_	+0.71	+0.78
-OH	+0.12	−0.37
-OCH_3_	+0.12	−0.22
-SH	+0.25	+0.15
-F	+0.34	+0.06
-Cl	+0.37	+0.23
-Br	+0.39	+0.23
-I	+0.35	+0.28
-H	0.00	0.00

**Table 4 materials-16-02148-t004:** Molecular structure and adsorption behavior of halogen-substituted N-heterocycles.

S No.	Chemical Structure/Abbreviation	Substituent	Adsorption Behavior	Ref.(s)
1	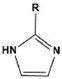	2-Cl-Imz: R=Cl2-Br-Imz: R=Br2-I-Imz: R=I	anodic-type inhibitor/Langmuir	[71]
2	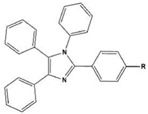	IM-Cl: R=ClIM-CH3:R=CH_3_	Mixed-type inhibitors/Langmuir	[54]
3	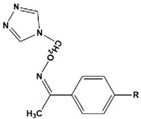	CATM: R=ClMATM: R=OCH_3_FATM: R=F	Mixed-type inhibitors/Langmuir	[72]
4	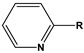	A: R=HB: R=BrC: R=Cl	-	[73]
5	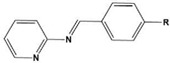	A: R=HB: R=CH_3_C: R=Cl	Mixed-type inhibitors/Langmuir	[46]
6	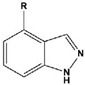	4-FIA: R=F4-CIA: R=Cl4-BIA: R=Br	Mixed-type inhibitors/Langmuir	[74]
7	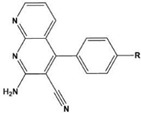	DHOP-F: R=FDHOP-Cl: R=ClDHOP-Br: R=Br	Mixed-type inhibitors/Langmuir	[75]

**Table 5 materials-16-02148-t005:** Statistical table of Gibbs free constants of corrosion inhibitors.

Corrosion Inhibitor	Metal	Corrosive Medium	Temperature	Adsorption Constant	Gibbs Free Energy of Adsorption (Kj/mol)	Ref.
[VMIM]I	X70 steel	0.5 M H_2_SO_4_	298	4865.2330 L/mol	−30.9851	[30]
[VPIM]I				5542.6228 L/mol	−31.3081	
[VBIM]I				7065.6398 L/mol	−31.9096	
[AEIM]Br	Copper	0.5 M H_2_SO_4_	298	0.53 × 10^5^ L/mol	−36.9	[31]
[ABIM]Br				2.17 × 10^5^ L/mol	−40.39	
[AHIM]Br				2.32 × 10^5^ L/mol	−40.56	
[AOIM]Br				2.50 × 10^5^ L/mol	−40.75	
PImC4	Aluminum alloy AA6061	1.0 M H_2_SO_4_	298	96 mmol^−1^	−21.3	[32]
PimC8				179 mmol^−1^	−22.8	
PimC12				350 mmol^−1^	−25.0	
A	Carbon steel	0.5 M HCl	298 K	4 L/mol	−12.3	[33]
B				333.3 L/mol	−24.3	
HBP	Carbon steel	1.0 M HCl	298 K	0.0342 L/mg	−32.86	[36]
OBP				0.0546 L/mg	−34.31	
DBP				0.1079 L/mg	−36.28	
8-Bpy	Carbon steel	1.0 M HCl	303 K	6.808 × 10^6^ L/mol	−48.93	[37]
10-Bpy				8.954 × 10^6^ L/mol	−49.61	
12-Bpy				3.630 × 10^6^ L/mol	−47.37	
APMC6	L X70 carbon steel	oilfield formation water	298 K	2.71 × 10^4^ L/mol	−	[38]
APMC12				3.91 × 10^4^ L/mol	−	
APMC18				5.35 × 10^4^ L/mol	−	
Tris-C1	Mild steel	1.0 M HCl	294 K	281 M^−1^	−23.6	[39]
Tris-C2				507 M^−1^	−25.0	
Tris-C3				1400 M^−1^	−27.5	
Tris-C4				1650 M^−1^	−27.9	
Tris-C5				4590 M^−1^	−36.1	
BTC_6_T	Carbon steel	1.0 M HCl	293 K	17.34 mM^−1^	−33.56	[40]
BTC_8_T				17.34 mM^−1^	−36.52	
IAH	Mild steel	0.5 M HCl	303 K	1.91 × 10^4^ M^−1^	−34.90	[41]
IBH				2.95 × 10^4^ M^−1^	−36.05	
IC-3	Mild steel	0.5 M HCl	303 K	-	−30.93	[27]
IC-5				-	−27.92	
IC-7				-	−26.44	
IC-11				-	−27.02	
IC-13				-	−27.92	
1-IM	Mild steel	0.5 M HCl	303 K	1.69 × 10^4^ L/mol	−34.65	[51]
2-IM				1.36 × 10^5^ L/mol	−39.67	
Comp. a	Carbon steel	0.3 M NaCl sat. Ca(OH)_2_	298 K	4.28 × 10^3^ L/mol	−30.7	[26]
ADTP1	Mild steel	1.0 M HCl	308	40.2 × 10^3^ M^−1^	−37.4	[45]
ADTP2				21.4 × 10^3^ M^−1^	−35.8	
ADTP3				10.2 × 10^3^ M^−1^	−33.9	
DAP-1	Mild steel	1.0 M HCl	308 K	5.8 × 10^4^ L/mol	−40.16	[46]
DAP-2				3.4 × 10^4^ L/mol	−39.19	
BI	Mild steel	1.0 M HCl	298 K	2 × 10^3^ M^−1^	−28.78	[58]
2-CH3-BI				2.5 × 10^3^ L/mol	−29.34	
2-SH-BI				10 × 10^3^ L/mol	−32.77	
AAC-1	Mild steel	1.0 M HCl	308 K	2.17× 10^4^ M^−1^	−35.87	[63]
AAC-2				2.63 × 10^4^ L/mol	−36.36	
AAC-3				3.32 × 10^4^ L/mol	−36.96	
HL1	C-steel	2 mol·L^−1^ HCl	303 K	4.39 × 10^5^ L/mol	−42.85	[21]
HL2				2.53 × 10^5^ L/mol	−41.46	
HL3				1.71 × 10^5^ L/mol	−40.48	
2-Cl-Imz	Mild steel	1.0 M HCl	-	1.2 × 10^3^ L/mol	−27.6	[73]
2-I-Imz				0.9 × 10^3^ L/mol	−26.7	
CATM	Mild steel	1.0 M HCl	298 K	1.74 × 10^4^ M^−1^	−34.15	[74]
FATM				1.75 × 10^4^ L/mol	−34.17	
4-FIA	Copper	3% NaCl	298 K	4.57 × 10^5^ L/mol	−42.24	[71]
4-CIA				7.19 × 10^5^ L/mol	−43.36	
4-BIA				4.93 × 10^5^ L/mol	−42.43	
DHOP-F	Mild steel	1.0 M HCl	298 K	1.89 × 10^4^ L/mol	−34.4	[76]
DHOP-Cl				2.20 × 10^4^ L/mol	−34.7	
DHOP-Br				2.52 × 10^4^ L/mol	−35.1	

## Data Availability

Data are contained within the article.

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
