# Peer review of "Revealing the Correlation between Molecular Structure and Corrosion Inhibition Characteristics of N-Heterocycles in Terms of Substituent Groups"

_materials, 2023, doi:10.3390/ma16062148_

Round 1

Reviewer 1 Report

See attached file.

Author Response

Thank you for sending us the reviewers’ comments concerning our manuscript entitled “Reveal the correlation between molecular structure and corrosion inhibition characteristics of N-heterocycles in terms of substituent groups” (ID: materials-2165961). Those comments are all valuable and very helpful for revising and improving the manuscript, as well as the important guiding significance to our research work. We have studied all the comments carefully and have made correction which we hope can meet with approval. Revised portion are marked in yellow and the main corrections with responds to the reviewer’s comments are provided as follows.

Please feel free to contact us with any questions and we are looking forward to your response.

With best wishes,

Yours sincerely,

Prof. Dr. Xiangqiong Zeng

Responds to the Reviewer 1:

  1. It is suggested to systematically inspect the related binding energies, ΔG, of the inhibitors with the metal surfaces, determined by experimental and theoretical methods. Such methods may reflect the state-of-the-art for the study of the inhibition corrosion. It will be useful to find the trends or correlation between ΔG and the inhibition corrosion performance. This analysis may involve also the effects of the transference of charge, between the inhibitor and the metal, which may increase the binding energy.

Response: Thank you for your careful review. We have summarized the Gibbs free energies of various corrosion inhibitors in the Table 5, and also summarized the law between the Gibbs free energies in the paper, which has been highlighted in yellow.

  1. For the protective layer-metal surface system it is suggested to report the trends of the shortest separations or equilibrium bond distances between the atoms or groups of the inhibitor molecules with the atom of the metal surface. Could the authors address the structural relaxation suffered by the adsorbed inhibitor molecules?

Response: Thank you very much for your valuable rimider. We reviewed the literature and found that there was relatively little literature on the minimum separation distance or equilibrium bond distance between surface corrosion inhibitors and metal adsorption bonds, which could not form a systematic comparison. That's a good point, and hopefully more data will support it.

  1. Though other metals are mentioned, the review is mainly focused on steel metal surfaces. It is suggested to address in more detail other metals such as bronze or other alloys.

Response: Thank the reviewers for the comment. For the series of compounds mentioned in the article, it is true that there are relatively few typical copper and other alloys. In order to increase the contrast, we added other metal compounds besides carbon steel, such as references 19,27,29,30,41,42,82.

  1. For a wider scope, the next reference should be added through the manuscript. Journal of Molecular Liquids, 2022, 363, 119826.

Response: Thank you very much for your valuable review. After careful reading of the manuscript (Journal of Molecular Liquids, 2022, 363, 119826). we found that the manuscript is very suitable for this article, and cited as reference 43.

Reviewer 2 Report

Manuscript ID: materials-2165961

Title: Reveal the correlation between molecular structure and corrosion inhibition characteristics of N-heterocycles in terms of substituent groups

This manuscript analyzed the influence of different substituents on the corrosion inhibition performance of N-heterocycles.

The manuscript is well written, the correlation between molecular structure and corrosion inhibition characteristics of N-heterocycles was comprehensively revealed, and their action mechanism was analyzed deeply.

Figures and tables support the authors' comments.

Some aspects of the paper need to be improved.

Introduction

That's quite a long introduction. I recommend shortening it by reformulation. Much information is repeated in other words.

Author Response

Thank you for sending us the reviewers’ comments concerning our manuscript entitled “Reveal the correlation between molecular structure and corrosion inhibition characteristics of N-heterocycles in terms of substituent groups” (ID: materials-2165961). Those comments are all valuable and very helpful for revising and improving the manuscript, as well as the important guiding significance to our research work. We have studied all the comments carefully and have made correction which we hope can meet with approval. Revised portion are marked in yellow and the main corrections with responds to the reviewer’s comments are provided as follows.

Please feel free to contact us with any questions and we are looking forward to your response.

With best wishes,

Yours sincerely,

Prof. Dr. Xiangqiong Zeng

Responds to the Reviewer 2:

  1. That's quite a long introduction. I recommend shortening it by reformulation.  Much information is repeated in other words.

Response: Thank you very much for your valuable rimider. We have cut the repetition of the preamble, and we have also revised some of the content, which has been marked.

Reviewer 3 Report

The manuscript makes a wide revision of N-heterocycle corrosion inhibitors, analyzing the influence of several factors (length of alkyl chain substituent, electron-donating, electron-withdrawing, presence of halogen atoms, etc.) on the performance of these compounds as well as mechanism types and toxicity. This is a good paper and will find interesting to many academics in the field. However, there are some info need to be included in the manuscript so that it will be more informative and increase the scientific value:

1.- Tables must be introduced in each section of the manuscript before being cited.

2.- The quality of Figure 2 should be improved. In addition, some other mechanisms presented in the manuscript should be in Figure form.

3.- Report or info on N-heterocycle corrosion inhibitors in pilot scale production needs to be highlighted.

4.- At the end of this review an outlook for the most prominent N-heterocyclic compounds should be given as well as the developing trend for them.

Author Response

Thank you for sending us the reviewers’ comments concerning our manuscript entitled “Reveal the correlation between molecular structure and corrosion inhibition characteristics of N-heterocycles in terms of substituent groups” (ID: materials-2165961). Those comments are all valuable and very helpful for revising and improving the manuscript, as well as the important guiding significance to our research work. We have studied all the comments carefully and have made correction which we hope can meet with approval. Revised portion are marked in yellow and the main corrections with responds to the reviewer’s comments are provided as follows.

Please feel free to contact us with any questions and we are looking forward to your response.

With best wishes,

Yours sincerely,

Prof. Dr. Xiangqiong Zeng

Responds to the Reviewer 3:

  1. Tables must be introduced in each section of the manuscript before being cited.

Response: Thanks to the reviewer for reminding us, we have introduced the table in the first paragraph of each section of the manuscript, highlighted in yellow

  1. The quality of Figure 2 should be improved. In addition, some other mechanisms presented in the manuscript should be in Figure form..

Response: Thanks for your suggestion, we have modified the Figure 5. At the same time, other mechanism diagrams are added in the manuscript, such as Figure 2,3,6 .

  1. Report or info on N-heterocycle corrosion inhibitors in pilot scale production needs to be highlighted.

Response: Thanks to the reviewer for reminding us. The N heterocyclic compounds investigated in this paper are still in the laboratory and theoretical stage, and have not reached the pilotscale production. We believe that more information will be added to this suggestion in future studies.

  1. At the end of this review an outlook for the most prominent N-heterocyclic compounds should be given as well as the developing trend for them.

Response: Thanks for the modification suggestions.We have added the prospect of the development trend of N heterocyclic compounds in “Conclusions and outlook” and marked with yellow.

Round 2

Reviewer 1 Report

The authors have properly attented my sggested improvement and corrections.

Thus, in my opinion this submitted research work has the merit for publication in the Journal of  Materials.